# A Review of Accelerated Long-Term Forgetting in Epilepsy

**DOI:** 10.3390/brainsci10120945

**Published:** 2020-12-07

**Authors:** Rūta Mameniškienė, Kristijonas Puteikis, Arminas Jasionis, Dalius Jatužis

**Affiliations:** 1Center for Neurology, Vilnius University, LT-08661 Vilnius, Lithuania; Arminas.jasionis@santa.lt (A.J.); dalius.jatuzis@santa.lt (D.J.); 2Faculty of Medicine, Vilnius University, LT-03101 Vilnius, Lithuania; kristijonas.puteikis@mf.stud.vu.lt

**Keywords:** accelerated long-term forgetting, epilepsy, memory impairment, transient epileptic amnesia, temporal lobe epilepsy

## Abstract

Accelerated long-term forgetting (ALF) is a memory disorder that manifests by a distinct pattern of normal memory for up to an hour after learning, but an increased rate of forgetting during the subsequent hours and days. The topic of ALF has gained much attention in group studies with epilepsy patients and the phenomenon has been shown to have contradictory associations with seizures, epileptiform activity, imaging data, sleep, and antiepileptic medication. The aim of this review was to explore how clinical and imaging data could help determine the topographic and physiological substrate of ALF, and what is the possible use of this information in the clinical setting. We have reviewed 51 group studies in English to provide a synthesis of the existing findings concerning ALF in epilepsy. Analysis of recently reported data among patients with temporal lobe epilepsy, transient epileptic amnesia, and generalized and extratemporal epilepsies provided further indication that ALF is likely a disorder of late memory consolidation. The spatial substrate of ALF might be located along the parts of the hippocampal–neocortical network and novel studies reveal the increasingly possible importance of damage in extrahippocampal sites. Further research is needed to explore the mechanisms of cellular impairment in ALF and to develop effective methods of care for patients with the disorder.

## 1. Introduction

Epilepsy is a disorder that manifests not only with seizures, but also affects the patients’ mood, behavior, cognition, and social interaction. The most common cognitive complaints in people with epilepsy are mental slowness, attention deficits, and memory difficulties [1,2]. The latter have gained special interest as neuropsychological testing in some epilepsy patients revealed normal recall and recognition at a normal delayed testing interval (e.g., 30–60 min), but deficits when testing for longer intervals (hours to days to weeks) [3]. This phenomenon was named accelerated long-term forgetting (ALF) and became a field of interest for further research of memory function in temporal lobe epilepsy (TLE), transient epileptic amnesia (TEA) and more recently—in extratemporal and genetic generalized epilepsies (ETE and GGE) [4,5,6,7]. ALF is detected by using methods that rely on testing patients on at least two separate occasions—one at a short delay (for example, 30 min) after having learnt word lists or visual material and one later (for example, after 1 week). Subjects that perform as well as controls after a short delay, but worse when tested later (hence, the “acceleration” in the rate of forgetting) are said to have ALF. Therefore, ALF is an objective result of repeated memory testing rather than a distinct disorder [3]. Subjective information provided by patients with ALF may include co-existing long-term memory deficits (e.g., patchy autobiographical memory), episodes of transient amnesia, or even no significant memory complaints [3,4,8].

Findings of impaired memory function at late delays are explained by the memory consolidation theory: long-term information integration to the neocortex is disrupted because of hippocampal lesions or seizure activity in the hippocampal–cortical network [9]. However, ALF itself and the research surrounding it are multifaceted. For instance, there are recent discussions whether the observations of normal short-term retrieval followed by a sudden drop in recollection results are biased. Elliot et al. noted various methodological problems with ALF studies, such as ceiling and floor effects, patient rehearsal, and control matching issues [10]. Further, the selection of long intervals between testing (e.g., second delayed recall at one week) does not directly prove a factual acceleration of forgetting. For instance, early divergence between two similar slopes of the time–memory relationship can remain non-significant only because of methodological constraints that hinder the detection of subtle deficits [11]. That is, ALF could delineate a continuous loss of memory with no acceleration—as in a form of modest hippocampal amnesia. Mayes et al. note that ALF may also represent an underlying slight memory deficit (difficult to detect through standard testing, but abnormal in essence) at first with increasing worsening after a certain point in time—a certain combination of “classic ALF” and mild hippocampal amnesia [12].

In this article, we seek to review recent findings of ALF in various types of seizure disorders by questioning (1) whether studies in epilepsy can help localize the physical substrate of ALF, (2) how ALF may be related to the disruption of memory consolidation, and (3) what could be the implications of these findings in the clinical setting. A narrative synthesis rather than a purely systematic approach is employed, considering that (1) studies of ALF in epilepsy tend to be very heterogeneous in their approach and methodology and (2) recent discussions reveal that there are theoretical grounds for ALF not being a singular disorder; both reasons making ALF a rather non-objective end-point [10,13,14].

## 2. Search Strategy and Results

We searched for articles investigating ALF in epilepsy patients by using a search string “accelerated” AND “forgetting” AND “epilepsy” to retrieve results from PubMed and Web of Science. To our knowledge, Fitzgerald et al. wrote the most recent review article with explicit emphasis on exploring ALF in relation to epilepsy through group studies—it was therefore added as an additional source [13]. The search string has been chosen because of the wide use of the term [10]. The procedure of article selection is presented in Figure 1. The inclusion criteria were (1) a group of patients with epilepsy or seizures tested for ALF through any selected material, (2) the procedure of ALF testing is described, and (3) the data for the selected recall and recognition points is presented and/or discussed. We omitted uncontrolled case reports and case series from the search protocol but included some notable examples in the discussion. For each article, we noted the type of epilepsy or seizures the patients had, whether they experienced seizures or were on antiepileptic medication. We presumed that ALF was present if the study group performed as well as or better than the control group at a short delay (e.g., up to an hour), but presented with a memory impairment at a longer delay. Systematic quality analysis by level of evidence was judged inappropriate. While methodological evaluation of ALF studies before 2014 has been performed based on specific criteria by Elliot et al., potential bias in other studies that are discussed is noted individually throughout this review [10]. 

Fifty-one article met the inclusion criteria. Sixteen studies explored ALF in patients with TEA and 21 included patients with TLE. Four studies examined patients with generalized epilepsies. Some reports included patient groups with different disorders: TLE and genetic generalized epilepsy (GGE)—3, TLE and extratemporal epilepsy (ETE)—5, various epilepsies—2. Seven studies (14%) did not observe ALF, suggesting the disorder may be frequent, but not ubiquitous among selected epilepsy patients. Methodological tendencies to produce positive results and publication bias should be considered as well.

The studies involving patients with TLE (both alone and including patients with ETE or GGE) are presented in Table 1. Reports exploring ALF in patients with TEA and other focal or generalized epilepsies are outlined in Table 2 and Table 3, accordingly.

## 3. ALF in Different Seizure Disorders

### 3.1. ALF in TLE—The Relation with Structural, Clinical, and EEG Variables

TLE is the most common form of focal epilepsies and is a major data source for ALF research [63]. Only one of the more recent studies (and 5 out of 29 in total) did not find ALF in a small sample of young patients [41]. Around half patients in a study of 36 participants with TLE or ETE have been positive for ALF with no relation to the statistical paradigm used [21]. Therefore, ALF in TLE might benefit not only memory research, but could be relevant in the routine clinical setting as well.

#### 3.1.1. The Hippocampus and ALF

The hippocampus as a central structure in memory formation exhibits pathologic changes in some forms of TLE and is evident to be involved in ALF [9,63]. Lah et al. show that ALF of verbal information in TLE is more common with structural hippocampal abnormalities, although not exclusively [29]. Other studies provide analogous associations of ALF and hippocampal abnormalities: worse recall of both autobiographic information and prose passages up to 24 h post-learning, worse verbal recall [23,31,32,35]. Correlating ALF and volumetric analysis (rather than binary presence or absence of lesions), however, appears to be less reliable [25]. ALF findings in non-verbal tasks are also not found to be associated with hippocampal variables [19,35]. Finally, there are indications that hippocampal damage determines the material-specific memory deficit (left damage for verbal, right for non-verbal tasks) during testing at short delays, but that the ALF pattern can be independent from the lateralization of hippocampal damage when investigated at longer delays (especially for verbal memory tasks) [22,25].

Novel studies of post-operative results provide additional clues of how hippocampal damage is related to ALF. Polat et al. demonstrated ALF in left TLE patients several years after amygdalohippocampectomy with subsequent freedom of seizures [18]. Visser et al. noted ALF in both left and right TLE in a group free of seizures [22]. Although resolution of ALF has also been reported post-operatively, one report consists of a case-study, while another study describes heterogeneous group level results with the authors’ emphasis on cautious interpretation (as some seizure-free TLE patients did experience ALF after surgery) [39,64]. In addition, the latter study employed story and visual tests that differed from the standard story or word recall testing in the reports mentioned previously. Thus, group-level findings indicate that ALF may persist with unilateral hippocampal loss and lack of overt seizures. By definition of ALF, patients in such cases should have normal early recall or recognition: a unilateral functioning hippocampus may provide encoding equivalent to preoperative and/or control levels. Some of this encoding can also be supported by posterior regions of the ipsilateral medial temporal lobe [65]. The second requirement to detect ALF is deficient long-term memory. It can be impaired because of pre-existing deficits in non-hippocampal regions, which remained unaffected by surgery. Therefore, neocortical (or hippocampal–cortical network) dysfunction or persisting subclinical seizure activity could explain the ALF found postoperatively [9,66,67]. Audrain et al. report a decrease in resting state functional connectivity between the anterior hippocampus ipsilateral to seizure onset and the temporal cortex contralateral to seizure onset, pointing towards the idea that contralateral spread of connectivity loss may predispose ALF by impairing alternative memory pathways [27,68]. In addition, inflammation of the amygdala in limbic encephalitis, but not hippocampal damage has been associated with ALF in one observational study [21]. This further strengthens to the notion that hippocampal dysfunction as such is not sufficient (and might not be central) to produce ALF.

#### 3.1.2. Material Specificity in ALF

Knowing that the dominant hemisphere is more associated with language function, an intuitive presumption would be that ALF of word or story material should be more prominent in left TLE and ALF of visuospatial data—in right TLE [69,70,71]. There is some evidence that this classic notion of hemisphere dominance may be adapted to the phenomenon of ALF. For example, left TLE has been found to predominate in several reports of ALF demonstrated by verbal memory testing [36,37]. Polat et al. reported ALF in seizure-free left TLE patients post-resection [18]. The authors speculate that bilateral hippocampal function may be required for early encoding, while structures of the intact left medial temporal lobe ensure consolidation of verbal information in the long term. It is unclear, however, how the post-operative deficit of hippocampal function influences the baseline rate of forgetting, and whether ALF in such studies can be distinguished as a phenomenon that is distinct from the consequences of the amygdaloidohippocampectomy [12,72,73]. In relation to ALF specific for the right hemisphere, Narayanan et al. argued that some visuospatial tasks (such as the Rey Osterrieth complex figure test) encompass a verbal component and therefore may not correlate with right side foci in ALF studies [36,74]. On the contrary, ALF of visual function-specific face recognition and route material has been found to be more prominent in right TLE [19,35]. Most of the recent studies, however, do not substantiate the lateralization hypothesis and report no significant changes in forgetting with regard to the side of seizure activity or hippocampal lesions [21,23,25,27,29,30,31,38]. Visser et al. note that methodological issues, such as different left/right categorizing may be in place [22]. For example, Blake et al. report verbal ALF in left TLE based on seizure activity, while Wilkinson et al. rely on hippocampal damage to define laterality [25,37]. The plastic reorganization of lateralized hemispheric function should also be considered to be a possible confounding factor [75,76]. Finally, even for the early delayed recall testing (such as 30 min after learning), the evidence for material-specificity is strong only for demonstrating association between the left hemisphere and verbal material [77,78,79]. In other cases, investigating material-specificity in ALF can be based on flawed presumptions that non-verbal memory is a construct opposite to verbal memory and that non-verbal tasks must be processed independently and in a lateralized way. In summary, while left TLE might be associated with verbal ALF and right TLE—with visuospatial ALF, additional imaging studies are required to determine, how cortical function localization and reorganization precisely influences ALF.

#### 3.1.3. Seizures and Epileptiform Activity

Seizure activity in TLE has contradictory associations with ALF. Seizures have been found to be the most important ALF predictor for autobiographical data in patients with focal seizures (including TLE) [32]. ALF of verbal, visual information has been linked to both seizures and interictal discharges [6,16]. Seizure frequency—both yearly and during the study period has been correlated to ALF as well [16,25]. In addition, patients who experienced complex partial and/or secondary generalized seizures performed significantly worse than patients with simple partial seizures [16]. Such findings, however, have not been widely replicated [5,30,31,35]. The postoperative studies reviewed previously show that absence of overt temporal seizures may not prevent ALF [18,22]. Subclinical interictal activity has been shown to be associated with ALF and may be a suspected cause of ALF-related disturbances in memory networks [6,32]. To the best of our knowledge, there are no studies examining patients who are both seizure-free and have observable interictal discharges. Considering the unexplored possibility of coexistent structural damage, methodological limitations (e.g., small sample size as in the study by Muhlert et al.) and the fact that seizure-free patients can present with ALF, it is difficult to determine, whether seizures influence ALF directly [5,10,25]. We believe that there is yet not enough evidence to prove that overt seizures or subclinical interictal activity are causative factors in ALF, especially considering that ALF is not epilepsy-specific (a point discussed later).

### 3.2. ALF in TEA—The Potential Importance of Subtle Cortical Dysfunction

#### 3.2.1. Clinical Characteristics of TEA

TEA may be understood as a TLE subtype comprised of a specific presentation of transient anterograde or retrograde amnesia episodes (mostly shorter than 60 min) with witnessed preservation of cognitive function during the event [4]. TEA patients are older, more often male and experience the attacks from several times a year to several times a day [4,44,46,80,81]. Amnesia on awakening, partial amnesia, topographic, and autobiographic memory impairment are characteristic of TEA [4,81,82]. Epilepsy-related criteria (introduced by Zeman et al. in 1998) are usually required to confirm the diagnosis: (1) epileptiform abnormalities on electroencephalography (EEG), (2) the concurrent onset of other clinical features of epilepsy (e.g., lip-smacking, olfactory hallucinations), and (3) a clear-cut response to anticonvulsant therapy [46,80,81]. However, not all of the criteria may be present in all suspected cases of TEA [80,83]. Antiepileptic drugs are reported to be very effective in resolving (in 73–96% of patients) or diminishing seizures [44,46]. ALF is present in around half of TEA patients and the memory impairment is more severe for those complaining of accelerated forgetting [46,82,84]. Long-term follow-up has shown that ALF disappears in some patients after the seizures have been controlled [53].

#### 3.2.2. Neuropsychological Testing Results

Just one of the 16 TEA studies reviewed did not report ALF [49]. The authors tested ALF in a picture recognition task and noted a constant rate of worsening recall in TEA patients. They argued that visual material recognition might be spared from accelerated forgetting in a sample with no patients having right hemisphere epileptiform discharges. Similarly, Atherton et al. noted only early encoding deficits for image recognition in TEA subjects, but no ALF [52]. Such constant deterioration of memory function due to acquisition deficits was not present in a study by Hoefeijzers et al. [51]. In fact, they demonstrated that by closely matching word learning conditions, ALF is still evident in patients with TEA. The same conclusion has been made by using delay intervals shorter than one week—ALF in a cued word recall test has been noted from 3 to 8 h post-learning [54]. Recall of real life events has been shown to be prone to ALF at a one day delay and to correlate with long-term recall of verbal material [57]. Further, a novel incidental story recall and recognition test revealed ALF that was more expressed for the recognition task [45]. Cued recall and recognition tasks prevent retrieval deficits to mask consolidation abnormalities. Therefore, results from such studies support the idea that ALF is the manifestation of a late consolidation impairment.

#### 3.2.3. Imaging Findings

Because the clinical presentation of TEA is more specific to memory dysfunction than in TLE, it is be reasonable to assume that both EEG and neuroimaging correlates in TEA might provide subtle insights into the pathogenesis of ALF. However, EEG results in TEA are nonspecific—unilateral (both left and right) or bilateral discharges may be recorded, and no epileptiform activity is found in up to 40% of cases [44,46]. Imaging studies are somewhat even less revealing than in TLE. Localized bilateral atrophy of structures like the hippocampus, the perirhinal and the orbitofrontal cortex has been found in patients with TEA [47,48]. However, the volumetric abnormalities, if reported at all, are subtle (e.g., 8% of total hippocampal volume) [44,50,51]. Additionally, while the volume of the mentioned structures is related to anterograde memory function, no correlation has been reported between the level of atrophy and ALF. Similarly, as in the case of TLE, these results indicate that the structural findings and EEG data in TEA currently provide no clear explanation of the pathogenetic mechanism of ALF. Curiously, olfactory hallucinations and olfactory impairment are found to be frequent in patients with TEA, pointing towards possible pathologic changes in structures, such as the piriform, entorhinal or orbitofrontal cortices [43]. Functional imaging is valuable by providing information about the metabolic changes in a discrete region and may serve to detect fine abnormalities. For example, a hypoactive hippocampus has been reported in TEA patients during visual material encoding [52]. Defects during this stage are hypothesized to render the memorizable information more susceptible to interference. These findings, however, encompassed just picture memory, for which ALF has not been found in the sample. In another study, hypometabolism of both the temporal and the prefrontal cortices have been demonstrated in TEA [44]. This provides objective evidence that, in patients with TEA, subtle metabolic deficits exist in both the hippocampus and the neocortex, and certain deficits in both information encoding and consolidation are plausible. Whether impaired hippocampal activity is enough to cause ALF in TEA, on the other hand, is unresolved.

In summary, findings from studies of TEA orient further research towards answering the question of how the impaired late consolidation in ALF could be directly associated with the pathological changes observable in imaging studies.

### 3.3. Focal Extratemporal Epilepsy and ALF

Focal extratemporal epilepsy in adults has been revealed to produce ALF in a story recall task after learning-to-criterion and to be present in individual cases while testing for verbal recall [23,31,38]. As mentioned previously, such uncued recall deficits may represent an impairment of information retrieval. This further adds to the idea that ALF is probably caused by problems in temporal and extratemporal neocortical sites and/or their connectivity with the hippocampus. Such localization would produce both the late (systems) consolidation deficits observable in TEA and TLE and the retrieval impairment in ETE.

### 3.4. ALF in GGE

In contrast to TLE and TEA, which encompass pathogenetic mechanisms in the memory structures of the temporal lobe, GGE has no such direct association. Nevertheless, both short and long-term memory function can generally be impaired in patients with GGE [85,86]. Further, the phenomenon of ALF has been demonstrated in children with GGE. While Davidson et al. found ALF in children with and without seizures, but associated the finding with poor initial learning, Gascoigne et al. showed that ALF is related to epilepsy severity and is still evident with controlled learning conditions in children who experience seizures [59,60]. In addition, there are indications that early-onset ALF in GGE does not subside over time-even with the reduction of seizures or the severity of epilepsy [7]. In this longitudinal study by Grayson-Collins et al., only recall was affected, thus revealing a potential retrieval deficit in the patients examined. While subclinical generalized discharges have been linked to poor long-term recall and ALF, the latter was absent in studies of adult GGE, including the report by Beilharz et al. [5,6,19,61]. The authors employed both the incomplete and complete learning paradigms, allowing comparison with the pediatric studies. A study by Muhlert et al. (evaluated being the best in terms of methodology by Elliot et al.) of adult patients with GGE did not report ALF either—forgetting of visual and verbal information was comparable to controls [5,10]. Overall, while generalized seizures do not seem to influence adult long-term retention on a level that would precipitate evident ALF, they may more likely impair retrieval in children.

## 4. ALF as a Late Consolidation Impairment

### 4.1. ALF in Non-Epileptic Disorders

Having reviewed recent group studies concerning ALF in epilepsy, we saw that as straightforward as ALF can seem in neuropsychological testing, it is a phenomenon associated with a broad spectrum of seizure disorders and diverse clinical, EEG and neuroimaging findings. To put these findings in perspective, it is noteworthy that ALF is not epilepsy-specific. For example, ALF is reported in subjects with autosomal dominant Alzheimer’s disease seven years before its estimated onset and those carrying the ε4 allele [87,88,89]. Further, there are accounts of recall deficits in children with traumatic brain injury (TBI), especially diffuse subcortical injury [90]. The authors considered retrieval difficulties to explain their findings and address a quite explicit inability to “pull information out” in other clinical situations concerning patients with TBI [91]. Minor stroke and transient ischemic attacks (TIA) have also been recently shown to provoke ALF [92]. This may indicate that ALF is a marker of non-disorder specific memory disruption and that epileptic activity is not essential for this condition.

### 4.2. Theoretical Grounds for Disrupted Long-Term Consolidation in ALF

The original paradigm of testing memory function at two distinct intervals (one short, one long-delay) is based on the “standard consolidation theory”. The latter poses that the hippocampus acts as a temporary hub for binding memory items into traces by “synaptic consolidation” to be later exported to neocortical locations through “systems consolidation” [93,94]. In this way, the encoded information ultimately will become independent from the hippocampus. This consolidation theory has been supplemented by alternative ideas, such as the "multiple trace theory", which argued that hippocampal–neocortical traces are strengthened over time, but both the hippocampus and the neocortex are required for memory storage and retrieval indefinitely [95]. Novel "unified" and “hippocampal-neocortical interactions” theories emphasize that both neocortical and hippocampal activation is required in initial cellular consolidation, after which the hippocampus forms axonal long-term projections with the neocortex or “indexes” neocortical memories for recall and reconsolidation [96,97,98]. Dudai et al. further note that neocortical sites are proved to act during encoding and engage in systems consolidation, which probably starts immediately at learning and may end even within hours (in case of an existing mental schema) [94]. Considering these theoretical developments, a continuum of consolidation processes (systems consolidation included) is in place from the very start of learning procedures. The disruption of the changes happening later in time, such as transcriptional modifications or synaptic growth, could therefore produce the “long-term forgetting” pattern, which is seen in ALF [99]. It is essential to say, however, that no fixed time period is required for the consolidation to proceed. The spectrum of timeframes, which make ALF evident, is therefore very variable—and became a central point of recent discussion concerning ALF. Cassel et al. reviewed the literature and argued that there is no evidence in ALF studies that a certain point in time exists, after which the acceleration of forgetting begins: it may in fact be the ceiling effects that mask the early acquisition deficits [35]. They went on to show that the forgetting patterns of visuospatial and verbal material in TLE patients is quite different in terms of timeframes, but that the acceleration in memory loss may become evident as early as 10 min post-learning. Similar accounts provide evidence of accelerated forgetting anywhere from minutes to hours after acquisition (rather than after 1 week) [54,57,100]. The biological logic of systems consolidation being a continuum and the existing evidence that ALF represents a progressive rather than a late divergent pattern of forgetting support the idea that the “acceleration” in ALF could be a product of statistical shortcomings [11]. This notion was included in a model of “ALF and mild hippocampal amnesia” by Mayes et al. [12]. The authors considered that the scientific bar to reject constant progression of forgetting by disproving even minimal short-term deficits might be very high. Therefore, a transition of slow-to-fast rate of forgetting could be a reasonable explanation of ALF. Due to the existing time overlap of early (cellular, synaptic) and late (systems) consolidation, such a model may be valid in the biological sense as well [93,94]. It would state that synaptic processes compensate for the disruption of systems consolidation at first, but a progressive deficit of long-term storage and/or retrieval emerges afterwards.

### 4.3. Is Seizure Activity or Structural Damage Essential to Cause ALF?

In this review, we considered the clinical, imaging and EEG correlates for ALF in TLE, TEA, ETE, and GGE. Prior to our analysis, Baker and Zeman proposed four possible mechanisms to explain the reasons underlying ALF in epilepsy: clinical or subclinical seizure activity, structural brain pathology, side effects of anti-convulsant drugs, and psychological mechanisms [3]. ALF, however, is not strongly associated with either of these clinical circumstances, except for some evidence linking ALF with subclinical interictal activity. We suggest that as much as the symptoms of long-term forgetting can be evident in the clinical setting, overt seizure activity or gross structural changes might overshadow the delicate pathology of ALF rather than explain it. For instance, it is unclear what part do hippocampal lesions play in the acceleration of forgetting; while the hippocampus probably acts in both synaptic and systems consolidation, its atrophy or injury seems to be more related to immediate anterograde deficits, not long-term forgetting. In fact, our review outlines many occasions where extrahippocampal sites (such as the amygdala, the prefrontal cortex) or connectivity problems are linked to observable ALF. As Helmstaedter et al. put it, “ALF seems to be a non-lateralized fronto-limbic dysfunction rather than a lateralized hippocampal dysfunction” [21]. There is also no solid evidence to consider seizures or subclinical interictal discharges as causing ALF—the underlying neuronal damage or synaptic reorganization, however, might unite the two [101]. This would in theory explain the accounts of ALF in non-epileptic disorders and the association of ALF with seizure severity in children GGE and adult TLE [3,16,60].

## 5. Translating the Research of ALF in Epilepsy to the Clinical Setting

### 5.1. Detecting ALF in Clinical Practice

The prevalence of ALF is uncertain, but memory complaints in epilepsy are frequent [102]. The benefits of searching for ALF routinely in patients with reported forgetfulness is questionable—it remains unclear, how such a finding would change the patient’s care strategy. It could nonetheless be important in substantiating the patient’s concerns by providing objective evidence of an existing memory disorder. Such validation of subjective complaints may relieve the patient’s distress by reassuring that ALF is not a functional disorder, but likely a consequence of epilepsy. However, ALF detection for both clinical and research purposes is not straightforward. The material employed to expose ALF varies in the type of information tested (verbal, story, picture, visuospatial design, route, real-life events etc.) while the paradigm employed during evaluation (recall, cued recall, and recognition) probably allows detection of distinct memory processes [10]. There are methodological recommendations to prevent bias during ALF testing and improve the patient experience [10,103]. These include testing for both verbal and visuospatial items, evaluating both recall and recognition, avoiding short-term memory influence (by prolonging the first testing interval) and ceiling, floor, and rehearsal effects. Further, new tests of both verbal and visuospatial memory are being explored for their potential use in ALF experiments [103]. Incidental memory tests may also be a practical solution as they would detect ALF for real-life events and be easy to effectuate in the clinical setting [45].

### 5.2. Treating Patients with ALF

The treatment of ALF in epilepsy relies on seizure control, which might resolve the long-term memory deficit. There is scant data of therapy options and their effectiveness to target ALF specifically. Individual TEA cases have been shown to benefit from carbamazepine [104,105]. Trials with lamotrigine and fluoxetine in TLE, however, did not improve ALF [34,100]. In contrast to prior opinions that polytherapy may influence long-term forgetting due to cognitive effects, it is unlikely to be associated with ALF [5,6,106]. Although antiepileptic drugs have been found to worsen memory at short delays, it is unclear, why they would affect late consolidation specifically [35]. As pointed out previously, while antiepileptic drugs are effective in TEA, the response could be worse in GGE and TLE. In addition, epilepsy surgery is employed in some TLE cases and effectively helps to control seizures, but the long-term memory deficit can persist [18,22]. While findings in some individual cases suggests that achieving seizure control may reduce the magnitude of ALF, there is as yet no systematic evidence to provide distinct recommendations to target ALF specifically through surgical or pharmacological methods.

### 5.3. Memory Rehabilitation

Alternative tactics to improve everyday information retention have been explored. For instance, the use of rehearsal and repetition, which is discouraged in clinical ALF trials, could benefit patients in means of overall recollection of memories [62,100]. Because recollection depends on initial acquisition, careful learning and/or treatment of attention deficits may simulate the learning-to-criterion paradigm and mitigate the negative effects of incomplete initial learning. Even wakeful resting after having learnt new data has been demonstrated to improve long-term recollection [107,108]. Del Felice et al. reviewed existing memory rehabilitation studies in patients with TLE and proposed that cognitive strategies, external mental aids, computerized training and non-invasive brain stimulation should be further explored to provide behavioral and mental techniques for reducing the burden of forgetfulness [109]. As recognition may be more intact in patients with ALF, Dubourg et al. suggest that a memory aid, such as the SenseCam (also used to measure ALF by Muhlert et al.), could provide patients with the required data to reach a certain memory threshold in recollecting real-life events [57,110].

Sleep is considered to be essential for memory consolidation by strengthening existing hippocampal–neocortical traces and protecting them from interference [9]. Earlier studies proposed that sleep is detrimental for ALF patients due to the epileptiform activity, which is often recorded during sleep [46,102]. Over time, however, new data proved a night’s sleep to be rather beneficial in ALF [40,55,56]. Further, a short nap has been found to aid consolidation for a design memory task in patients with ALF [6]. Despite this, the theory of interictal activity interference during consolidation in sleep may not be wrong per se. Patients with TEA benefit from sleep less than controls and the amount of slow wave sleep in patients is correlated negatively with the retention benefits [56]. Concerning the existing data, patients experiencing ALF could be encouraged to take regular naps.

## 6. Conclusions

This section summarizes the ideas expressed in the review according to the authors’ initial aims.

### 6.1. Do Memory Studies in Patients with Epilepsy Reveal a Possible Localization of ALF?

Many of the studies reviewed report a direct association between hippocampal damage and ALF. However, the significance of such findings is partly undermined by several aspects: (1) ALF may persist after hippocampal surgery; (2) hippocampal damage or atrophy is neither required, nor sufficient to detect ALF; and (3) alternative regions of memory pathways also exhibit pathological changes and may better explain the distinctive pattern of memory decay. The latter is presumably caused by the disruption of long-term consolidation, which relies on extrahippocampal sites. Thus, current evidence supports localizing the substrate of ALF in both the hippocampus and distinct other structures, including the hippocampal–neocortical network, the neocortex itself and the amygdala.

### 6.2. ALF as a Reflection of Impaired Late Memory Consolidation

By matching initial learning, using cued recall or recognition tests, preventing ceiling; and floor effects, it has been substantiated that ALF is a distinct phenomenon that differs from typical amnesia because of preserved early memory formation. Further, ALF is seen in various patient groups; thus, the etiology of any hypothesized neuronal deficit is not disorder-specific. In this way, ALF might be the expression of a common neurocognitive defect that can be determined by several distinct pathological mechanisms (e.g., defective transcriptional or synaptic network changes), but result in a similar pattern of deficient late consolidation and long-term memory. In epilepsy, for instance, seizures and interictal epileptiform activity might either damage neurons directly or interfere in their interaction during late consolidation. However, there is little data to support clear causative mechanisms of ALF by either seizure activity, structural damage, or cellular alterations.

### 6.3. Future Directions of ALF Research and Its Detection in Clinical Practice

As it is yet unknown whether ALF is primarily caused by structural damage, functional changes or both, the genesis of ALF requires further investigation. While animal studies could explore the cellular changes in ALF, imaging studies, connectome analysis are further required to reveal the neural network diversification in humans. It would be also of interest to discover the prevalence of ALF among patients throughout the neurological disease spectrum. If the phenomenon were frequent as a non-specific finding of neural damage, this would have direct implications. For example, ALF could become a biomarker in asymptomatic patients and be examined for its prognostic value, especially considering progressive neurodegenerative diseases or prediction of the course of epilepsy. Finally, interventions (e.g., cognitive, behavioral, or computerized methods) that reduce the burden of ALF could be explored further. They may simultaneously provide new insights into the memory patterns in ALF and benefit the patient.

### 6.4. Final Remarks

After having reviewed current group studies concerning ALF in epilepsy, we have the impression that ALF research is expanding rapidly both in quantity and methodological techniques. This variability in testing material, measured outcomes (i.e., cued, uncued recall or recognition) and study samples (e.g., different seizure disorders, age groups, and clinical context) limits study interpretation and direct comparison. This review was intended to present a broad outlook on ALF in epilepsy without quantitative consideration to study quality, thus limiting its generalizability.

## Figures and Tables

**Figure 1 brainsci-10-00945-f001:**
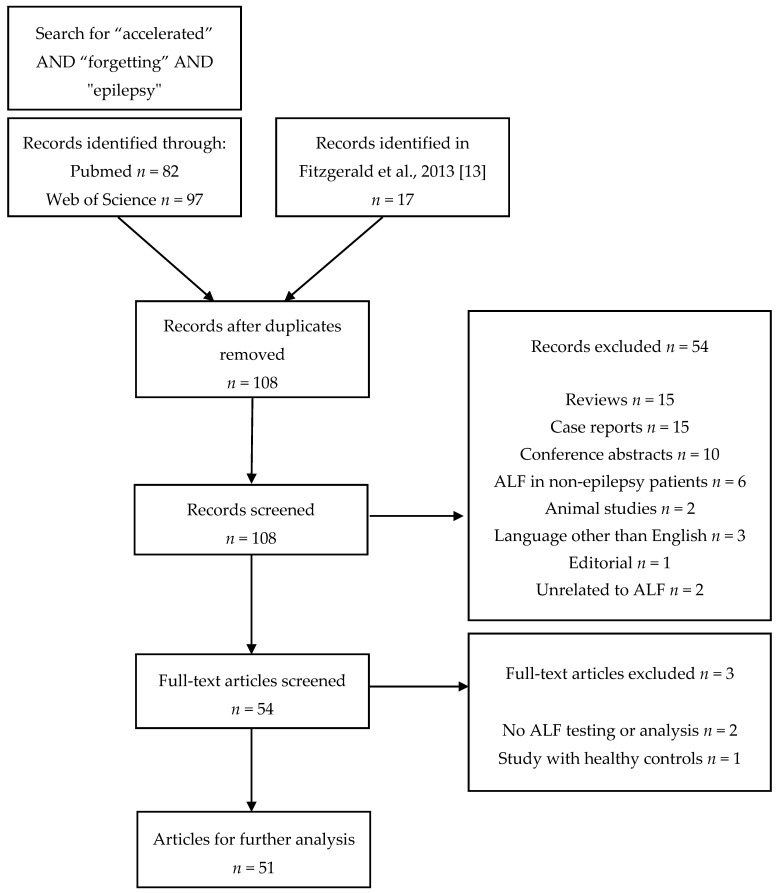
The search protocol employed in the review.

**Table 1 brainsci-10-00945-t001:** Group studies of patients with temporal lobe epilepsy (TLE). Ordered in descending number of TLE patients involved. Information that was lacking or is not applicable is presented as blank cells. AEDs—antiepileptic drugs, CVLT-C—California Verbal Learning Test-Children’s Version, DCS-R—Diagnosticum für Cerebralschädigung-Revised, ETE—extratemporal epilepsy, GASE—Global Assessment of the Severity of Epilepsy, GGE—genetic generalized epilepsy, HVLT-R—Hopkins Verbal Learning Test-Revised, IGE—idiopathic generalized epilepsy, L-TLE—left temporal lobe epilepsy, M/F—male/female, RAVLT—Rey Auditory Verbal Learning Test, ROCF—Rey-Osterrieth Complex Figure, R-TLE—right temporal lobe epilepsy, SD—standard deviation, SRT—selective reminding test, sz—seizure, TEA—transient epileptic amnesia, VLMT—Verbal Learning Memory Test, WMS-R-LM—Wechsler Memory Scale-Revised-Logical Memory.

Reference	Epilepsy Types	Controls (M/F)	Patients (M/F)	Lateralization (M/F)	Patient Age (Mean, SD or Range)	GASE Rating (SD)	AEDs (Mean, SD if Indicated)	Experiences Seizures? Frequency, SD (if Indicated)	Material	ALF Detected?	Delay Points	Comment
Djordjevic et al., 2011 [15]	TLE	11/8	44/46	R-TLE: 14/30 L-TLE: 30/16	R-TLE: 36.8 L-TLE: 33.5				Story recall	No	30 min, 24 h	
Mameniskiene, Jatuzis, Kaubrys, & Budrys, 2006 [16]	TLE	19/40	29/41		33.3 (9.5)		36 on monotherapy 34 on polytherapy	Yes, 6.6 (5.6) sz/month	RAVLT ROCFT Story recall	Yes (all tests)	30 min, 4 weeks	ALF worse with frequent seizures
Helmstaedter, Hauff, & Elger, 1998 [17]	TLE	11/10	27/28	R-TLE: 27 L-TLE: 28	26.9		Yes	Yes (but not during the past 8 h)	VLMT DCS-R	Yes (both tests)	30 min, 1 week	
Polat et al., 2020 [18]	TLE	10/14	23/28	R-TLE: 10/15 L-TLE: 13/13	R-TLE 36.44 (8.30) L-TLE 37.15 (9.47)			No (Engel class I)	VLMT WMS-R-LM	Yes (seizure free L-TLE)	30 min, 1 week, 6 weeks	Follow-up years after epilepsy surgery
Bengner et al., 2006 [19]	TLE and GGE	6/6	20/36	R-TLE: 11/13 L-TLE: 7/13 IGE: 2/10	39.2 (11.8)		Yes	Yes (9 during delay)	Face recognition	Yes	2 min, 24 h	Face recognition is worse in R-TLE
Bell, Fine, Dow, Seidenberg, & Hermann, 2005 [20]	TLE	22/27	14/28	R-TLE: 9/11 L-TLE: 5/17	R-TLE: 40.0 (9.8) L-TLE: 34.0 (13.0)		Yes, 1.9 (0.75) (R-TLE), 1.7 (0.70) (L-TLE) on average	Yes, monthly	Word list SRT recall Design SRT recall	No	30 min, 24 h	
Helmstaedter, Winter, Melzer, Lohmann, & Witt, 2019 [21]	TLE and ETE	65/89	16/20	Temporal: 32 (72% LE+) Extratemporal: 4 (0% LE+) Left:15 Right:14 Bilateral: 7	43 (17)		Yes	Yes, LE+ 2.6 (1.7) sz/month, LE- 5.9 (6.4) sz/month	VLMT	Yes	30 min, 1 week	ALF in 31–67% patients
Visser et al., 2019 [22]	TLE	14/16	15/15	R-TLE: 17 L-TLE: 13	41.3 (19–62)		4 on monotherapy 26 on polytherapy	Yes: 14 No: 16	RAVLT	Yes (both R/L-TLE)	30 min, 1 week	Post antero-temporal lobectomy with amygdalo-hippocampectomy (16 Engel class I, 4 Engel class II, 10 Engel class III). ALF in the seizure-free group.
Miller, Mothakunnel, Flanagan, Nikpour, & Thayer, 2017 [23]	TLE and ETE	29/31	22/22	Temporal: 27 Extratemporal: 14 Temporal and extratemporal: 3 Left:22 Right:21 Bilateral: 1	40.7 (13.4)		Yes	No: 20 Yes: 24 (>1 sz/y)	RAVLT Aggie	Yes (especially RAVLT)	30 min, 1 week	23% of patients with ALF. Both temporal and extratemporal. ALF frequent with hippocampal lesions.
Giovagnoli, Casazza, & Avanzini, 1995 [24]	TLE	13/12	14/14	R-TLE: 5/7 L-TLE: 9/7	38 (11.6)		Yes	No (past 6 months): 7 Yes: 21	Design recall	No	1 h, 1 day, 3 days, 6 days, 13 days	
Wilkinson et al., 2012 [25]	TLE	22	27	R-TLE: 12 L-TLE: 15 (based on hippocampal sclerosis)	R-TLE: 38.67 (8.07) L-TLE: 34.80 (10.13)		Yes, 25 on polytherapy	Yes, uncontrolled	Story recall ROCFT	Yes (both tests)	1 h, 6 weeks	Different ALF patterns, depending on lateralization
Bell, 2006 [26]	TLE	8/17	10/15	R-TLE: 6 L-TLE: 11 Bilateral: 2 Uncertain: 6	39 (10)		Yes: 23 (1.5 (1.0)) No: 2	Yes: 11 Yearly seizures/seizure-free: 14	WMS-III-LM	No	30 min, 2 weeks	
Audrain & McAndrews, 2019 [27]	TLE	12/12	14/9	R-TLE: 9 L-TLE: 12 Bilateral: 2	37.17 (12.35)			Over 72 h delay: Yes: 8 No: 8 N/a: 7	Digital object-scene pair recognition	Yes	15 min, 90 min, 6 h, 16 h, 72 h	Indications of greater forgetting after 90 min
Gascoigne et al., 2019 [28]	IGE and TLE	27/31	IGE 10/10 TLE 10/13	R-TLE: 6 L-TLE: 15 n/a: 2	11.3 (3.3)	IGE 2.1 (1.7) TLE 2.6 (1.2)	IGE: 17 on monotherapy, 3 on polytherapy TLE: 2 no AEDs, 14 on monotherapy, 7 on polytherapy	Yes (at least one 2 years prior)	CVLT-C	Yes	2 min, 30 min, 1 week	ALF associated with behavioral problems
Lah, Mohamed, Thayer, Miller, & Diamond, 2014 [29]	TLE	13/14	9/14	R-TLE: 13 (7 with hippocampal lesions) L-TLE: 10 (5 with hippocampal lesions)	44.8 (13.4)		3 no AEDs 11 on monotherapy 9 on polytherapy	No: 5 < 1 sz/month: 8 >1 sz/month: 10	HVLT-R	Yes	30 min, 1 day, 1 week	ALF worse with hippocampal abnormalities
Gascoigne et al., 2014 [30]	TLE	27/31	10/13	R-TLE: 2/4 L-TLE: 6/9 Unconfirmed: 2	12.5 (2.8)	2.6 (1.2)	2 no AEDs 14 on monotherapy 7 on polytherapy	Yes	CVLT-C Visual memory (Design location)	Yes (CVLT-C)	2 min, 30 min, 1 week	ALF unrelated to epilepsy variables
Ricci, Mohamed, Savage, & Miller, 2015 [31]	TLE and ETE	29	32	TLE: R-TLE: 12 L-TLE: 5 Bilateral: 1 Unconfirmed: 3 ETE: Right: 3 Left: 3 Bilateral: 1 Unconfirmed: 4	39.0 (11.8)		Yes, 1.7 (1.1) on average	Yes, 18.6 (22.7) sz/year	Prose passage recall and recognition	Yes	30 min, 24 h, 4 days	TLE, hippocampal lesions—worse recall between 30 min and 24 h ETE—greater decay between 24 h and 4 days
Ricci, Mohamed, Savage, Boserio, & Miller, 2015 [32]	TLE and ETE	29	32	TLE: 21 (12 with hippocampal lesions) ETE: 11	39.0 (11.8)		Yes	Yes	Autobiographical experience recall and recognition	Yes	30 min, 24 h, 4 days	Hippocampal lesions—worse recall between 30 min and 24 h Seizures, discharges-greater decay between 24 h and 4 days
Martin et al., 1991 [33]	TLE	6/15	10/11	R-TLE: 4/4 L-TLE: 6/7	31 (7.5)			Yes	Word list SRT recall	Yes	30 min, 24 h	
Barkas et al., 2012 [34]	TLE	6/6	10/13	R-TLE: 7/6 L-TLE: 3/7	48				Virtual water maze	Yes	3–6 weeks	Both hippocampal sclerosis and surgery groups demonstrate ALF
Cassel, Morris, Koutroumanidis, & Kopelman, 2016 [35]	TLE	9/9	9/9	R-TLE: 6 L-TLE: 6 Bilateral:4 n/a: 2	39.3 (9.8)		1 no AEDs 9 on monotherapy 8 on polytherapy	Yes, 5 during participation	Story task Route video recall	Yes (story task)	30 s, 10 min, 1 day, 1 week	Faster route forgetting in R-TLE
Narayanan et al., 2012 [36]	TLE	3/14	6/9	R-TLE: 2/4 L-TLE: 4/4	R-TLE: 34.5 (12.21) L-TLE: 32.88 (9.09)			Yes	RAVLT ROCFT Labyrinth maze Autobiographic events	Yes (RAVLT)	30 min, 4 weeks	
Blake, Wroe, Breen, & Mccarthy, 2000 [37]	TLE and others	6/10	7/14 (14 TLE)	R-TLE: 5 L-TLE: 9	R-TLE: 36.67 (9.41) L-TLE: 33.20 (10.94)		Yes, 2.05 (0.86) on average	Yes	Story recall and recognition	Yes	30 min, 8 weeks	
Muhlert et al., 2011 [5]	TLE and GGE	TLE: 3/7 GGE: 7/8	TLE: 4/10 GGE: 6/8	EEG: Left: 2 Right: 2 Bilateral: 2 n/a: 8	TLE: 46.4 (11) GGE: 31.6 (14.6)		TLE: 2 no AEDs 6 on monotherapy 6 on polytherapy GGE: 1 no AEDs 8 on monotherapy 5 on polytherapy	Yes, 3.8 sz/month (TLE), 5.4 sz/month (GGE)	Visual scene recall and recognition Story recall and recognition Spatial discrimination Descriptive recall	Yes (TLE, visual scene recall, descriptive recall, story recognition)	40 s, 30 min, 3 weeks	ALF not related to seizure variables. ALF in TLE, not GGE.
Miller, Flanagan, Mothakunnel, Mohamed, & Thayer, 2015 [38]	TLE and ETE	28/32	7/8	TLE: R-TLE: 3 L-TLE: 6 ETE: Left: 2 Right: 2 Bilateral: 1 Unconfirmed: 1	37			Yes	RAVLT WMS-IV-LM Aggie figures	Yes (all tests, RAVLT more sensitive)	30 min, 1 week	ALF predominance in TLE. ALF irrespective of laterality.
Evans, Elliott, Reynders, & Isaac, 2014 [39]	TLE	12/13	3/4	R-TLE: 2/2 L-TLE: 1/2	39.71 (15.77)		2 on monotherapy 5 on polytherapy	Yes	Visual scene recall and recognition Story recall and recognition	Yes (both tests)	30 min, 1 week	ALF tested pre- and post-operatively: ALF improvement after resection
Deak, Stickgold, Pietras, Nelson, & Bubrick, 2011 [40]	TLE	9	7	R-TLE: 1 L-TLE: 5 Bilateral: 1	44.0		2 on monotherapy 5 on polytherapy	Yes, 1 (during testing)	Word list SRT recall Finger tapping motor sequence	Yes (word test)	30 min, 12 h, 24 h	Indication of sleep being protective in ALF
Contador, Sánchez, Kopelman, & González de la Aleja, 2017 [41]	TLE (one patient with TLE+TEA)	4/6	2/3		24.40 (4.09)		2 on monotherapy 3 on polytherapy	Yes (4.2 sz/year)	Cued story recall Cued visuospatial route recall	No	30 s, 10 min, 1 day, 1 week	
Tramoni et al., 2011 [42]	TLE and TEA	15	4/1	Left: 1 Right: 2 Bilateral: 2	42.6 (9.3)		Yes, on monotherapy	No (past 1 year)	Story recall and recognition Virtual route recall and recognition Real route recall and recognition Ecological episode recall and recognition Single item recognition Recall and recognition of new facts	Yes (story, route, episode tasks)	1 h, 6 weeks	

**Table 2 brainsci-10-00945-t002:** Group studies involving patients with transient epileptic amnesia (TEA). Ordered in descending number of TEA patients involved. Information that was lacking or is not applicable is presented as blank cells. AEDs—antiepileptic drugs, GKMDT—Graham–Kendall Memory for Designs test, M/F—male/female, RAVLT—Rey Auditory Verbal Learning Test, ROCF—Rey-Osterrieth Complex Figure, SD—standard deviation, WMS-III-LM—Wechsler Memory Scale-Third Edition-Logical Memory.

Reference	Epilepsy Type	Controls (M/F)	Patients (M/F)	Patient Age (SD)	AEDs	Experiences Seizures? Mean, SD if Indicated	Material	ALF Detected?	Delay Points	Comment
Savage, Butler, Milton, Han, & Zeman, 2017 [43]	TEA	38/12	42/13	70.59 (8.14)		Yes, mean number of attacks 17.85 (18.05)	RAVLT	Yes	30 min, 1 week	Olfactory hallucination and reduction in olfactory function more frequent in TEA.
Mosbah et al., 2014 [44]	TEA	15	18/12		1 no AEDs 24 on monotherapy 5 on polytherapy	Yes	WMS-III-LM ROCF Single items recognition	Yes (WMS-III-LM, ROCF)	1 h, 6 weeks	Neurometabolic correlates for ALF in the medial temporal lobe.
Hoefeijzers, Zeman, Della Sala, & Dewar, 2019 [45]	TEA	14/18	20/7	66.44 (9.48)	27 on monotherapy	No (past 6 months)	Incidental story recall and recognition	Yes	30 min, 1 week	
Butler et al., 2007 [46]	TEA	24	34/16 (24 tested for ALF)	68 (8.7)	Yes, 48 on monotherapy	No (between sessions)	RAVLT GKMDT	Yes (both tests)	30 min, 1 week, 3 weeks	
Butler et al., 2013 [47]	TEA	20	28/13 (22 tested for ALF)	67.7 (8.9)	Yes, on monotherapy	No (past 6 months)	Word, prose, and design recall	Yes	30 min, 1 week, 3 weeks	No significant structural MRI correlates with ALF
Butler et al., 2009 [48]	TEA	20	28/13 (22 tested for ALF)	67.7 (8.9)	Yes, on monotherapy	No (past 6 months)	Word, prose, and design recall	Yes	30 min, 1 week, 3 weeks	Bilateral hippocampal volumes smaller in patients
Dewar, Hoefeijzers, Zeman, Butler, & Della Sala, 2015 [49]	TEA	8/8	12/4	69.63 (6.33)	Yes, on monotherapy	No (past 6 months)	Picture recognition test	No	5 min, 2.5 h, 7.5 h, 24 h, 1 week	In TEA patients, picture recognition impaired after 5 min with no acceleration in rate afterwards
Butler, Kapur, Zeman, Weller, & Connelly, 2012 [50]	TEA	8/12	12/10	66.4 (8.8)	Yes	No (during study)	RAVLT	Yes	30 min, 1 week	Hippocampal volumetry not related to ALF
Hoefeijzers, Dewar, Della Sala, Zeman, & Butler, 2013 [51]	TEA	7/11	9/8	65.47 (8.79)	Yes, on monotherapy	No (past 6 months)	Word recall (reanalysis of Butler et al., 2007 [46])	Yes	30 min, 1 week, 3 weeks	Support for ALF as a consolidation issue
Atherton, Filippini, Zeman, Nobre, & Butler, 2019 [52]	TEA	10/5	12/3	67.73 (1.63)	Yes, on monotherapy	No (past 6 months)	Digital image recognition RAVLT	Yes (RAVLT)	30 min, 1 week	Significant fMRI results (hypoactive left hippocampus prior forgetting)
Savage et al., 2019 [53]	TEA	7/5	9/5	78.3 (7.0)	1 no AEDs 13 on monotherapy	At T2: No (stable): 9 Yes: 5	RAVLT Story recall GKMDT	Yes (story recall, GKMDT)	30 min, 1 week	10-year follow-up (4 converters ALF to no ALF, but 0 converters from no ALF to ALF)
Hoefeijzers, Dewar, Sala, Butler, & Zeman, 2015 [54]	TEA	8/8	10/1	69.82 (5.60)	Yes, on monotherapy	No (past 6 months)	4 Word Lists, recall and recognition	Yes	30 min, 3 h, 8 h, 24 h, 1 week	Evidence of ALF from 3 to 8 h post-learning
Atherton, Nobre, Zeman, & Butler, 2014 [55] and Atherton et al., 2016 [56]	TEA	7/5	10/1	67.73 (1.63)	Yes, on monotherapy	No (past 6 months)	Word pairs association	Yes	30 min, 12 h	Sleep promotes memory retention in ALF
Muhlert, Milton, Butler, Kapur, & Zeman, 2010 [57]	TEA	1/10	10/1	68.6 (9.9)	Yes, on monotherapy	No (past 4 months)	Word recall Real-life event SenseCam test	Yes (both tests)	30 min, 1 day, 1 week, 3 weeks	
Manes, Graham, Zeman, De Luján Calcagno, & Hodges, 2005 [58]	TEA	7	6/1	57 (8.1)	Yes, on monotherapy	No	Story recall and recognition Drawing recall and recognition	Yes (story tests)	30 min, 6 weeks	

**Table 3 brainsci-10-00945-t003:** Studies including only patients with genetic generalized epilepsy (GGE) or other epilepsy types (temporal lobe epilepsy and transient epileptic amnesia excluded). Ordered in descending number of patients involved. Information that was lacking or is not applicable is presented as blank cells. AEDs-antiepileptic drugs, ALF—accelerated long-term forgetting, CVLT-C—California Verbal Learning Test-Children’s Version, CVLT-III—California Verbal Learning Test-Third Edition, GASE—Global Assessment of the Severity of Epilepsy, M/F—male/female, SD—standard deviation, WMS-IV-LM—Wechsler Memory Scale-Fourth Edition-Logical Memory.

Reference	Epilepsy Type	Controls (M/F)	Patients (M/F)	Patient age (Mean, SD, or Range)	GASE Rating (SD)	AEDs	Experiences Seizures? Frequency, SD (if Indicated)	Material	ALF Detected?	Delay Points Tested	Comment
Fitzgerald, Thayer, Mohamed, & Miller, 2013 [6]	Various Normal EEG: 18 Focal: 10 Generalized: 5	15	34	Normal EEG: 36.89 (10.47) Focal: 41.8 (14.60) Generalized: 32.60 (14.54)		4 no AEDs 17 on monotherapy 12 on polytherapy	Yes	Word Memory Test Design Memory Test	Yes (both tests)	30 min, 24 h, 4 days	No relation with seizure variables. Information better retained with naps.
Davidson, Dorris, O’Regan, & Zuberi, 2007 [59]	GGE	21	7/14	11.5 (8–16)		4 no AEDs 15 on monotherapy 2 on polytherapy	Yes: 16 No: 5	Children’s memory scale: Stories and Dot location subtests	Yes (stories task)	30 min, 1 week	Poor initial learning
Gascoigne et al., 2012 [60]	GGE	20/21	10/10	10.76 (2.47)	2.17 (1.69)	18 on monotherapy 2 on polytherapy	Yes (at least one 2 years prior)	CVLT-C	Yes	2 min, 30 min, 1 week	Epilepsy severity related to ALF
Grayson-Collins et al., 2019 [7]	GGE	11/18	11/7	10.90 (2.53)	1.68 (1.08)	7 no AEDs 7 on monotherapy 4 on polytherapy	At T2: No: 7 Yes: 11	CVLT-C	Yes	2 min, 30 min, 1 week	No improvement in ALF after seizure reduction (7.53 (2.43) years of follow-up)
Beilharz, Thayer, Nikpour, & Lah, 2020 [61]	GGE	2/12	4/12	42.93 (13.09)	3.31 (1.18)	5 on monotherapy 9 on polytherapy	n/a, last seizure at 4.51 (8.60) years	CVLT-III WMS-IV-LM	No	2 min, 30 min, 1 week	
Ricci, Wong, Nikpour, & Miller, 2019 [62]	Symptoms of epilepsy	10/0	3/0	66.7 (5.7)		2 no AEDs 1 on monotherapy	No	Story recall	Yes	30 min, 24 h, 1 week, 2 weeks, 4 weeks	Testing ALF patients after a 2 weeks delay and early rehearsal improved recall up to 1 week

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
