# Peer review of "A Review of Accelerated Long-Term Forgetting in Epilepsy"

_brainsci, 2020, doi:10.3390/brainsci10120945_

Round 1

Reviewer 1 Report

Summary of this paper is below.

ALF is a memory disorder that is different from the short-term memory but lasts hours and days. The ALF with epilepsy patients is known, however the symptoms  are not associated with seizures, epileptiform activity, imaging data, sleep, and antiepileptic medication. The purpose of this systematic review was that how clinical and imaging data could help diagnose ALF.

Many of the studies reviewed still report the association between hippocampal damage and ALF, however, a gradual shift can be observed towards relating ALF to extrahippocampal sites.

This is an important review article.

I have some minor comments.

L127, “Engel class I” is out of the date. I can understand the paper that you cited used the term "E-Class I”. However, as this is the latest publication, I guess readers might feel the term as outdated. You just say “freedom of seizures” or add explanation that E-Class I is equivalent to ILAE outcome class 1.

L151-152, “Polat et al. reported ALF in seizure-free left TLE patients post-resection. This might be post-operative memory disturbance. There are two case reports (Front Neurol. 2019 Jun 12;10:620. doi: 10.3389/fneur.2019.00620., Nat Med. (2017) 23:678–80. doi: 10.1038/nm.4330) that the only invasive electrodes could detect the seizures that caused ALFs. I guess they just did not show clinical manifestation. However, subclinical seizures can induce the ALFs. These papers also explain the relationship between hippocampus and neocortex that might support your theory.

L200, For the Zeman’s definition of the TEA, we know that the three conditions are required. However, I guess we do not necessarily follow the classical definition (Psychogeriatrics. 2017 Nov;17(6):491-492. doi: 10.1111/psyg.12249. Epub 2017 Mar 27.not always).  

Author Response

We thank the reviewer for his effort and time to comment on our manuscript. We provide point-by-point answers below and indicate the changes made. Besides, the “track changes” function has been turned-on to view all modifications in the manuscript file (the line numbers we indicate refer to lines in “simple markup” or “no markup” viewing options).

Reviewer 1

Summary of this paper is below.

ALF is a memory disorder that is different from the short-term memory but lasts hours and days. The ALF with epilepsy patients is known, however the symptoms  are not associated with seizures, epileptiform activity, imaging data, sleep, and antiepileptic medication. The purpose of this systematic review was that how clinical and imaging data could help diagnose ALF.

Many of the studies reviewed still report the association between hippocampal damage and ALF, however, a gradual shift can be observed towards relating ALF to extrahippocampal sites.

This is an important review article.

I have some minor comments.

L127, “Engel class I” is out of the date. I can understand the paper that you cited used the term "E-Class I”. However, as this is the latest publication, I guess readers might feel the term as outdated. You just say “freedom of seizures” or add explanation that E-Class I is equivalent to ILAE outcome class 1.

We agree, now L133 “amygdalohippocampectomy with Engel class I outcome” is changed to “amygdalohippocampectomy with subsequent freedom of seizures”. However, we keep Engel Class classification terms in the tables as they refer to direct data provided by the authors whose publications we refer to.

L151-152, “Polat et al. reported ALF in seizure-free left TLE patients post-resection. This might be post-operative memory disturbance.

We agree, the possibility of post-operative memory disturbance has been mentioned in L164-165: “whether ALF in such studies can be distinguished as a phenomenon that is distinct from the consequences of the amygdaloidohippocampectomy”

There are two case reports (Front Neurol. 2019 Jun 12;10:620. doi: 10.3389/fneur.2019.00620., Nat Med. (2017) 23:678–80. doi: 10.1038/nm.4330) that the only invasive electrodes could detect the seizures that caused ALFs. I guess they just did not show clinical manifestation. However, subclinical seizures can induce the ALFs. These papers also explain the relationship between hippocampus and neocortex that might support your theory.

We thank for the references included. They help to substantiate our claim concerning subclinical epileptiform activity, which has already been mentioned in L145-147: “Therefore, neocortical (or hippocampal-cortical network) dysfunction or persisting subclinical seizure activity could explain the ALF found postoperatively”. We added the case reports suggested as references for this statement.

L200, For the Zeman’s definition of the TEA, we know that the three conditions are required. However, I guess we do not necessarily follow the classical definition (Psychogeriatrics. 2017 Nov;17(6):491-492. doi: 10.1111/psyg.12249. Epub 2017 Mar 27.not always). 

We agree that recording all three criteria is best to confirm TEA. We also agree that there may be necessary to have only one or two criteria to consider making the diagnosis. For instance, Zeman et al. wrote in 1998: “Of these three criteria, the first two are the strongest. Two of the patients in this study (cases 2 and 3) satisfy all three criteria; four others (cases 2, 8, 9, and 10) satisfy two; cases 4, 5, 6, and 7 satisfy one criterion each. We have included the third group, in which the diagnosis is least certain, on the hypothesis that the clinical features of their attacks are identical to those in the first two groups and reflect, therefore, a common mechanism of transient amnesia” and also separated the criteria by “either…or” in the Abstract “This evidence was provided by either (a) wake or sleep EEG, or (b) the co-occurrence of other seizure types (if their roughly concurrent onset or close association with episodes of tran- sient amnesia suggested a connection), or (c) a clear cut response to anticonvulsant therapy, or by a combination of these three factors”.

L200-204:

We changed: One or more of the three epilepsy-related criteria (introduced by Zeman et al. in 1998) are usually required to confirm the diagnosis: (1) epileptiform abnormalities on electroencephalography (EEG), (2) the concurrent onset of other clinical features of epilepsy (e.g. lip-smacking, olfactory hallucinations) and (3) a clear-cut response to anticonvulsant therapy.

To: Epilepsy-related criteria (introduced by Zeman et al. in 1998) are usually required to confirm the diagnosis: (1) epileptiform abnormalities on electroencephalography (EEG), (2) the concurrent onset of other clinical features of epilepsy (e.g. lip-smacking, olfactory hallucinations) and (3) a clear-cut response to anticonvulsant therapy. However, not all of the criteria may be present in all suspected cases of TEA.

(we added as reference the publication you provided after the last sentence: Ukai, K., & Watanabe, M. (2017). Transient epileptic amnesia without epileptic seizures: proposal of a new entity. Psychogeriatrics, 17(6), 491–492. https://doi.org/10.1111/psyg.12249).

Reviewer 2 Report

Summary:

The manuscript is a retrospective chart review of long-term accelerated forgetting from 51 published papers. The review concludes that ALF is likely a disorder of late memory consolidation and may be related to related to extra-hippocampal as well as hippocampal brain regions.

Comments:

The review appears to be a comprehensive review of a phenomenon that is not often discussed. In part for this reason, the review may benefit from a more careful definition of ALF. For example, how precisely does ALF differ from long-term memory loss, what is mean by “accelerated”, and how is it defined in studies. Indeed, many epilepsy patients report memory difficulties. It wasn’t quite clear to me how ALF differs from these typical complaints or how the information presented in the review would improve understanding or treatment of these patients. One additional consideration is whether ALF at all relates to anti-seizure medications.

Finally, it might be helpful if it were a little clear how the objectives of the review (listed L55-57) were met or answered by the review. Perhaps improved organization of the Results/Discussion and/or a concluded section with brief conclusions would help accomplish this. I find the paragraphs in the Results/Discussion section to be dense/wordy and somewhat confusing in their organization.

Minor comments:

Abstract: “very contradictory associations” might be re-phrased or clarified. Typically, either something is or isn’t “contradictory”.

Tables: need re-formatting; words are currently split mid-syllable decreasing readability

Author Response

We thank the reviewer for his effort and time to comment on our manuscript. We provide point-by-point answers below and indicate the changes made. Besides, the “track changes” function has been turned-on to view all modifications in the manuscript file (the line numbers we indicate refer to lines in “simple markup” or “no markup” viewing options).

Reviewer 2

Summary:

The manuscript is a retrospective chart review of long-term accelerated forgetting from 51 published papers. The review concludes that ALF is likely a disorder of late memory consolidation and may be related to related to extra-hippocampal as well as hippocampal brain regions.

Comments:

The review appears to be a comprehensive review of a phenomenon that is not often discussed. In part for this reason, the review may benefit from a more careful definition of ALF. For example, how precisely does ALF differ from long-term memory loss, what is mean by “accelerated”, and how is it defined in studies. Indeed, many epilepsy patients report memory difficulties. It wasn’t quite clear to me how ALF differs from these typical complaints or how the information presented in the review would improve understanding or treatment of these patients.

We thank you for the comment and agree that ALF might be quite difficult to distinguish from long-term memory loss and typical memory complaints. We added an additional explanation of the definition of ALF. We state that ALF is an objective clinical finding, not a memory complaint or a distinct disorder. Further, we added a sentence that comparison with a matched control is required to determine, whether one has ALF. The latter sentence includes the explanation of “accelerated” as this refers to a decay of memory that is faster than the one in the control group (during the interval between testing). Finally, we emphasized that the finding of ALF may be unrelated to co-occurring clinical findings or complaints as they differ greatly between patients with detectable ALF.

L36-43: ALF is detected by using methods that rely on testing patients on at least two separate occasions – one at a short delay (for example, 30 min) after having learnt word lists or visual material and one later (for example, after 1 week). Subjects that perform as well as controls after a short delay, but worse when tested later (hence the “acceleration” in the rate of forgetting) are said to have ALF. Therefore, ALF is an objective result of repeated memory testing rather than a distinct disorder [3]. Subjective information provided by patients with ALF may include co-existing long-term memory deficits (e.g., patchy autobiographical memory), episodes of transient amnesia or even no significant memory complaints [3,4,8].

Considering the question of  “how the information presented in the review would improve understanding or treatment of these patients”, we added an additional sentence in the section regarding ALF in the clinical setting:

L364-365: Such validation of subjective complaints may relieve the patient’s distress by reassuring that ALF is not a functional disorder, but likely a consequence of epilepsy.

Regarding the implications of ALF research on its detection and treatment, subsections (5.1. Detecting ALF in clinical practice, 5.2. Treating patients with ALF, 5.3. Memory rehabilitation) are now included to better find these relevant parts of the review.

These aspects are now clarified in the Conclusion section as well:

L349-444 If the phenomenon were frequent as a non-specific finding of neural damage, this would have direct implications. For example, ALF could become a biomarker in asymptomatic patients and be examined for its prognostic value, especially considering progressive neurodegenerative diseases or prediction of the course of epilepsy. Finally, interventions (e.g. cognitive, behavioral, computerized methods) that reduce the burden of ALF could be explored further. They may simultaneously provide new insights into the memory patterns in ALF and benefit the patient.

One additional consideration is whether ALF at all relates to anti-seizure medications.

We believe the relationship with AEDs has already been addressed in L377-389 (both regarding treatment options and the fact that AEDs are unlikely to cause ALF). This section now has a separate subheading “5.2. Treating patients with ALF”.

Finally, it might be helpful if it were a little clear how the objectives of the review (listed L55-57) were met or answered by the review.Perhaps improved organization of the Results/Discussion and/or a concluded section with brief conclusions would help accomplish this. I find the paragraphs in the Results/Discussion section to be dense/wordy and somewhat confusing in their organization.

Thank you for addressing the problems of wordiness, density and issues with organization. To improve the layout of the review, we reorganized several paragraphs by switching their places and by introducing new subheadings (please also refer to the changes presented in the table below). The conclusions were also reorganized: now each objective from the Introduction is considered individually under separate subheadings (6.1.-6.3.).

Lines (current document, “simple markup”

Previous text

Current text

L9-11

Accelerated long-term forgetting (ALF) is a memory disorder that manifests itself by a distinct pattern of normal short-term memory, but increased rate of forgetting during the subsequent hours and days

Accelerated long-term forgetting (ALF) is a memory disorder that manifests by a distinct pattern of normal memory for up to an hour after learning, but an increased rate of forgetting during the subsequent hours and days

L44-46

Findings of impaired function with late delays are being explained by the memory consolidation theory: long-term information integration to the neocortex is disrupted due to hippocampal lesions or seizure activity in the hippocampal-cortical network and this could explain the memory pattern in ALF

Findings of impaired memory function at late delays are explained by the memory consolidation theory: long-term information integration to the neocortex is disrupted because of hippocampal lesions or seizure activity in the hippocampal-cortical network

L49

…biased per se.

…biased.

L80-84

We presumed that ALF was present if the study group performed as well as or better than the control group at a short delay (i.e. up to an hour), but presented with a memory impairment at a longer delay. Systematic quality analysis by level of evidence was judged to be inappropriate: while methodological evaluation of ALF studies before 2014 has been performed earlier based on specific criteria by Elliot et al., potential bias in other studies that are discussed is noted individually throughout this review [9].

We presumed that ALF was present if the study group performed as well as or better than the control group at a short delay (e.g., up to an hour), but presented with a memory impairment at a longer delay. Systematic quality analysis by level of evidence was judged inappropriate. While methodological evaluation of ALF studies before 2014 has been performed based on specific criteria by Elliot et al., potential bias in other studies that are discussed is noted individually throughout this review [10].

L94

The studies considering patients with TLE…

The studies involving patients with TLE…

L119-120

Hippocampus as a central structure in memory formation both exhibits pathologic changes in some forms of TLE and is evident to be somehow involved in ALF

The hippocampus as a central structure in memory formation exhibits pathologic changes in some forms of TLE and is evident to be involved in ALF

L137-L139

In addition, the latter study employed story and visual tests that differed from the standardized story or word recall testing in the reports mentioned previously.

Besides, the latter study employed story and visual tests that differed from the standardized story or word recall testing in the reports mentioned previously.

L140-153, L161, L161-164, L189, L192, L200

[minor changes, spliting of complex sentences, additional explanatory sentences, reduces wordiness]

L223-232

[the paragraph has been moved to be under an individual subsection]

L235, L40-241, L244, L246, L248-249

[minor changes, spliting of complex sentences, additional explanatory sentences, reduces wordiness]

L255-256

How this is related to ALF, on the other hand, is a separate question.

Whether impaired hippocampal activity is enough to cause ALF in TEA, on the other hand, is unresolved.

L313-314

Maybe structural injury in cerebrovascular or traumatic disorders was less intuitively relatable to a direct dysfunction in memory circuits – especially with the co-occuring plethora of other urgent symptoms, which are of greater priority in both research and clinical practice. If so, finding ALF in TEA and TLE early on could have been more determined by the dominance of memory complaints and lack of their explanation (due to normal function at short delays) in seizure disorders than by ALF being selectively present in epilepsy.

This may indicate that ALF is a marker of non-disorder specific memory disruption and that epileptic activity is not essential for this condition.

L333-338

The authors considered that the scientific bar to reject constant progression of forgetting (by disproving even minimal short-term deficits) might be very high, thus, a transition of slow-to-fast rate of forgetting could be a reasonable explanation of both ALF and the biases of its research. Due to the existing time overlap of early (cellular, synaptic) and late (systems) consolidation, such a model is valid in the biological sense as well [75,76]. Disruption of systems consolidation would be somewhat compensated by the synaptic processes for some time, and a progressive deficit of long-term storage and/or retrieval would reveal itself afterwards.

The authors considered that the scientific bar to reject constant progression of forgetting by disproving even minimal short-term deficits might be very high. Therefore, a transition of slow-to-fast rate of forgetting could be a reasonable explanation of ALF. Due to the existing time overlap of early (cellular, synaptic) and late (systems) consolidation, such a model may be valid in the biological sense as well [93,94]. It would state that synaptic processes compensate for the disruption of systems consolidation at first, but a progressive deficit of long-term storage and/or retrieval emerges afterwards.

Minor comments:

Abstract: “very contradictory associations” might be re-phrased or clarified. Typically, either something is or isn’t “contradictory”.

We agree. Line 12: “the phenomenon has been shown to have very contradictory associations” now changed to “the phenomenon has been shown to have contradictory associations”

Tables: need re-formatting; words are currently split mid-syllable decreasing readability

We have re-formatted the tables by reducing the text size as this improves readability and prevents word-splitting. References of all articles in the tables are now added not only as text, but as functioning number references (and can be found in the list at the end of the manuscript). Minor typographical errors in the tables have also been corrected (including one rather inconsequential change in the “ALF found” section in Table 1: Bell et al., 2005: ALF was mistakenly attributed as a finding in this study).

Round 2

Reviewer 2 Report

Thank you for your attention to the revisions. The authors' have satisfied my concerns.